# Towards LLMs as Operational Copilots for Fusion Reactors

Viraj Mehta[1], Joe Abbate[2], Allen M. Wang[3], Andy Rothstein[2,4], Ian Char[1], Jeff Schneider[1], Egemen Kolemen[2,4], Cristina Rea[3], and Darren T. Garnier[3]

[1]Carnegie Mellon University, Pittsburgh, PA, USA
[2]Princeton Plasma Physics Laboratory, Princeton, NJ, USA
[3]MIT Plasma Science and Fusion Center, Cambridge, MA, USA
[4]Department of Mechanical and Aerospace Engineering, Princeton University, Princeton, NJ, USA

## Abstract

The tokamak is one of the most promising approaches for achieving nuclear fusion as an energy source. As such, there are many tokamaks with rich experimental histories and datasets. While the quantitative data generated by tokamaks is invaluable, tokamak operations also generate another, often underutilized data stream: text logs written by experimental operators. In this work, we leverage these extensive text logs by employing Retrieval-Augmented Generation (RAG) with state-of-the-art large language models (LLMs) to create a prototype "copilot". Instances of this copilot were created using text logs from the fusion experiments DIII-D and Alcator C-Mod and deployed for researchers to use. In this paper, we report on the datasets and methodology used to create this "copilot", along with its performance on three use cases: 1) semantic search of experiments, 2) assisting with device-specific operations, and 3) answering general tokamak questions. In the first two use cases, we observe clear advantages over base LLMs and simple keyword search. However, a survey of researchers indicates that the RAG system does not clearly outperform the base model for general questions related to tokamak operations.

## 1 Introduction

If realized as an energy source, nuclear fusion could forever change the way that we power humanity. Not only does it have a relatively abundant fuel source, but unlike nuclear fission, it has no risk of catastrophic meltdown. Perhaps the most promising device for nuclear fusion is the tokamak: a toroidal machine that magnetically confines a plasma. Despite all the promise, there are still important open questions that stand between humanity and economical tokamak power plants.

To answer these questions, nuclear fusion scientists have access to several experimental devices that can be used to test hypotheses or try new control techniques. For example, scientists competitively bid for experiment sessions on the DIII-D tokamak. These sessions may last anywhere from three hours to a multi-day campaign, and during this time the device can be run approximately four times an hour (each run is known as a "shot"). Although the leaders of these experiments must carefully make a "shot plan" ahead of time, they must also be reactive to what is happening in the experiment as well as flexible with the current limitations of the device, which evolve throughout every day of operations. There is substantial cognitive load on these "session leaders": they must manage a team of operators and technicians who handle the subsystems and overall function of the machine, observe and process incoming data about the previous experiments, and decide what configuration changes to effect for the next shot, all within a tight time budget and serious safety constraints. When making

NeurIPS 2023 AI for Science Workshop.

these quick adjustments, it is often helpful to draw on the rich source of information provided by the logs from past experiments. These logs contain textual information about what was observed in past shots, problems that arose on the device and how operators addressed them, and experiments that were tried on the device. Session leaders have very little time to sort through these logs while leading an experiment; even if a valuable nugget of information is contained within, it is unlikely they would be able to unearth it under the real-time constraints of the setting.

Additionally, there have been hundreds of fusion reactors for which there is little or no institutional memory, such as the Tokamak Fusion Test Reactor (TFTR) [Hawryluk et al., 1991]. Finding and utilizing shot logs for old devices could help uncover information otherwise lost to history. Another loss of information happens as scientists retire. The machine National Spherical Torus Experiment Upgrade (NSTX-U) Menard et al. [2017] has been offline for nearly a decade and many former operators have left, their knowledge lost with them. Making sure historical information is not lost will be an important role as NSTX-U returns to operation.

To address these challenges, we draw on the recent successes of Retrieval-Augmented Generation (RAG)[Lewis et al., 2020], which exploits large language models (LLMs) such as GPT [Brown et al., 2020, OpenAI, 2023] and Llama [Touvron et al., 2023] to answer questions using concrete references to tokamak experimental logs. RAG "copilots" were created for two fusion experiments: one for DIII-D, an active fusion experiment based in San Diego, California and operated by General Atomics Luxon et al. [2005]; and one for Alcator C-Mod Hutchinson et al. [1994], a fusion experiment at the MIT Plasma Science and Fusion Center (PSFC) which was retired in 2016. These two instances are stylized as "ChatDIII-D" and "ChatCMod" respectively. ChatDIII-D was deployed as a Discord bot [1] serving operations at General Atomics, and ChatCMod was deployed as a web interface accessible to researchers on the MIT PSFC private network.

We begin by describing our database in section 2, and then in section 3 we describe the RAG algorithm implemented along with LLM tools used. Then in section 4, we demonstrate the performance of the co-pilot in answering queries submitted by fusion scientists We conclude in section 5 by giving our vision for the future of such tools to better assist operators.

## 2 Text Database

Many fusion experiments, including DIII-D and Alcator C-Mod, have digital text log entries written by experiment operators that describe what happened during each shot [Fredian and Stillerman, 2006]. These text logs are invaluable sources of information about difficult-to-observe factors that influence the shot including, but not limited to, hardware and software anomalies, operational changes, and unusual experimental procedures such as the intentional injection of impurities. In addition, the text logs also record many relevant or interesting phenomena. In the following subsections, we describe the text databases of Alcator C-Mod and DIII-D used in this work.

### 2.1 Alcator C-Mod Text Logs

We used a 95MB database of text logs from 44,261 Alcator C-Mod shots spanning operations from 1993 to 2016. Each shot log consists of multiple "entries" with the format:

```
SHOT NUMBER
ENTRY TOPIC: # Name of topic.
ENTRY USER:  # Name of user writing the log.
ENTRY TEXT:  # User text.
```

The names of users were anonymized. We treat each entry as a "document" for embedding and retrieval. We also excluded entries under the "rf_monitor" topic from the database as they contain large volumes of raw sensor data that took up large amounts of the context window without seeming to offer any value.

---

[1]Discord is used at DIII-D to facilitate communications among a distributed research team

## 2.2 DIII-D Text Logs

We used a database of 83,882 DIII-D shots spanning operations going from 1986 to 2023. There are two different types of log entries: individual shot entries (734MB) and run-day summaries (142MB). An individual shot entry will have:

```
SHOT NUMBER
USER ROLE:    # Role of user making log entry.
ENTRY USER:   # Name of user writing the log.
ENTRY TEXT:   # User text.
```

These entries tend to contain information specific to changes in diagnostics, heating sources such as the neutral beams or electron cyclotron heating, or notes on the specific experimental goals in that shot.

In addition to specific shot notes, there are summary notes for each experimental run day consisting of anywhere from 15 to 30 shots depending on the length of the experiment. These summary logs contain higher level information such as which parts of the experiment were a success, if there were repeated machine problems plaguing performance, and any high performing "trophy" shots that are worth noting. While the summary notes will have less information on an individual shot basis, they have significantly more information about recurring problems and interesting fixes that were found during the experimental day.

## 3 Methods

In this section, we first describe the RAG algorithm implemented before describing the language and embedding models used along with the data restrictions motivating the choices. The models used are standard, off-the-shelf models which are fine-tuned and prompt-engineered to better suit the task of a tokamak operator copilot.

### 3.1 Retrieval Augmented Generation (RAG)

We leverage retrieval augmented generation (RAG) [Lewis et al., 2020], which takes advantage of both the general reasoning and language skills of the LLM and the domain knowledge encoded in our dataset of experimental summaries and shot log entries in order to provide relevant and appropriate context to the copilot prototype. This technique aims to address the issue that LLMs are typically trained on general datasets but are able to reason in-context when presented with additional information relevant to the task. Here, we give a brief summary of the technique. First, we embed all potentially relevant documents into a vector space. Next, we embed the user's query into the same space and find $k$ approximate nearest neighbors. We include the documents corresponding to the nearest neighbor embeddings in a prompt which emphasizes the user's query. The LLM then generates text based on a response to the query as well as the included documents in context.

In order to be able to reference information from a wide range of documents, we "chunk" our documents by splitting them into overlapping segments of a moderate length. Due to context length limitations we used a chunk size of 300 tokens for run summary documents and a chunk size of 400 tokens for shot log documents. We also overlap each adjacent chunk by 60 tokens in order to ensure that no information is unduly split across chunks. We then embed each chunk using an embedding model and store them in a Chroma Contributors [2023] vector database for convenience and efficient approximate retrieval.

### 3.2 Language and Embedding Models Used

Due to data policies on the DIII-D text database, we were restricted to using open source models that were run locally. Specifically for our experiments, we used Sentence Transformers [Reimers and Gurevych, 2019] for embeddings and llama-2-70b Touvron et al. [2023] for text generation. Alcator C-Mod, on the other hand, is a retired fusion experiment which allowed us to use the OpenAI API under their new data policy which promises that user data will remain private, will not be used for training, and will be deleted within 30 days. Specifically, we used the "text-embedding-ada-002" model for embedding and GPT-4 for text generation OpenAI [2023].

# 4    Evaluating the co-pilot

In this section, we evaluate the RAG co-pilot on a number of scenarios that could be useful to fusion scientists. The first focuses on finding previous experiments (Section 4.1), the second focuses on an example scenario in which device-specific operational assistance is given (Section 4.2), and in the last section we compare answers from the RAG copilot vs raw GPT-4 on general tokamak questions (Section 4.3).

## 4.1    Semantic Search of Experiments

Fusion researchers often have a need for identifying shots with a particular phenomenon or event. In fact, 12 out of the 33 questions asked during the initial deployment of the "ChatCMod" website, and three out of the seven questions asked during initial deployment of the "ChatD3D" server (Appendix A), were questions requesting shots with given characteristics. Historically, users would complete this task by either identifying and asking a subject matter expert, or performing a keyword search on the logbook. However, the former requires waiting potentially days for an answer and the latter is incapable of identifying shot logs that are semantically relevant but do not have the exact search strings, requiring extra effort to construct a reasonable regular expression.

One case where our implementation provided an advantage over simple keyword search is when a ChatCMod user asked the question: "What shots have fast rotating MHD". In response, the chatbot utilized logged frequency numbers to distinguish "fast" versus "slow" MHD activity, something that would not have been possible with simple keyword search.

```
Question:
What shots have fast rotating MHD?

Answer:
Looking at the shot notes, a few instances of fast rotating MHD
    activity emerges:

1. Shot 1101203015: The entry indicates MHD activity at a
    frequency of over 650 kHz, which is attributed to fast ion−
    driven Toroidal Alfven Eigenmodes (TAEs). This frequency range
     suggests fast rotating activity.
2. Shot 1090918006: A 40kHz rotating mode starts up between 1 and
    1.05 seconds. Though not as high as the first example, 40kHz
    still represents relatively fast rotation.
3. Shot 1110316033: A higher frequency mode slows down from 60 to
    40 kHz during the high−power phase after 1.3 seconds. That
     still counts as fast rotating since the frequencies are up to
    60kHz before slowing down.
4. Shot 1120221017: The MHD in this shot is classified as being "
    shorter−lived" compared to another, with high−frequency
    activity that grows stronger after 0.78 seconds. This suggests
     faster rotating activity.

The above information suggests these shots had fast rotating MHD
    activity. However, it's important to acknowledge that "fast"
    is relative, and these examples are based on the highest
    identified frequency values in the provided records.

The encountered shots that mention fast rotating MHD activity are
    1101203015, 1090918006, 1110316033, and 1120221017.
```

For example, it discovered shot number "1120918024" which mentions a "bright rotating mode at 55kHz...". While the word "fast" doesn't appear in the text, the retrieval step and language model interpretation step both picked up on the high kHz number.

```
1120918024_7: CMOD Shot Number: 1120918024
ENTRY TOPIC: MHD
ENTRY USER: {redacted}
ENTRY TEXT:
<pre>
Bright rotating mode at 55kHz from .7 to .95 sec, along with
weaker shadows at ~45 and 38kHz. Sawteeth look distorted during
this period, possible locked mode.
</pre>
```

## 4.2 Device-Specific Operational Assistance

In addition to semantic search, the tool can field questions that are specific to an individual machines' control systems. For example, the plasma in a tokamak sits within a hollow toroidal chamber. A cross-section is shown in figure Figure 1. The distance between the plasma and the walls of the container is an important parameter. Moving the plasma closer to the walls stabilizes the plasma; but moving the hot plasma too close increases the temperature of the wall, which can reduce plasma performance and possibly damage hardware. One of the authors was once told to reduce the "outer wall gap" (i.e. distance from the outer plasma edge to the outer wall) in the tokamak's Plasma Control System (PCS). However, while the DIII-D-specific control system has settings for shape parameters like the plasma radius "Rp" and the *inner* wall gap, there is no explicit setting for the *outer* wall gap. An experienced operator was found and queried to learn that decreasing the outer wall gap by 2 units is usually done by increasing the major radius by 1 unit while holding the inner wall gap (and all other settings) fixed. This is a rule of thumb operators generally use based on their past experience and knowledge of the specific shape control GUI implementation, based on the intuition shown in Figure 1. To reduce cognitive load on experienced operators and teach more thoroughly from previous examples, naive questions like this can be fielded by the co-pilot:

```
Question:
How do you decrease the outer wall gap in the PCS?

Answer:
 Based on the information provided in the shot notes, it appears
     that there are a few different ways to decrease the outer
      wall gap in the PCS. Here are some possible methods that have
       been used in the past:

1. Increase the Rp program: By increasing the Rp program, the
     outer gap can be reduced. This was done in shot 165965, where
     the Rp program was increased by 1.0 cm (from 9.5 to 7.5
     cm) to reduce the outer gap from 9.5 cm to 7.5 cm.
.
.
.
```

Note that in the corresponding log there is no appearance explicitly of "outer wall gap". The retriever correctly interpreted "outer gap" as relevant, as shown in the abridged version of the shot notes:

```
shot 165965: ... Repeat co-NBI case with smaller outer gap of 7.5
     cm... Reduce the outer gap by 2.0 cm (from 9.5 to 7.5 cm) by
     increasing Rp by 1.0 cm (constant inner gap)... set vertices:
     Rp/ShotStart/Shape ...
```

The co-pilot correctly identifies the "Rp" ("major Radius of Plasma") parameter in the control system as the relevant vertex to change. It also brings attention to the important factor of 2 between the Rp and the outer gap change. However, it does not emphasize that the inner gap is maintained constant (the intuitive reason for the factor). Additionally, in red is a clear factual misinterpretation of the retrieved notes. Finally, the generator continued on with factually incorrect proposals (not shown here) based on other shot notes. Nonetheless, by reading the corresponding shot notes highlighted by the generator the user can quickly and easily reference the context necessary to understand how to

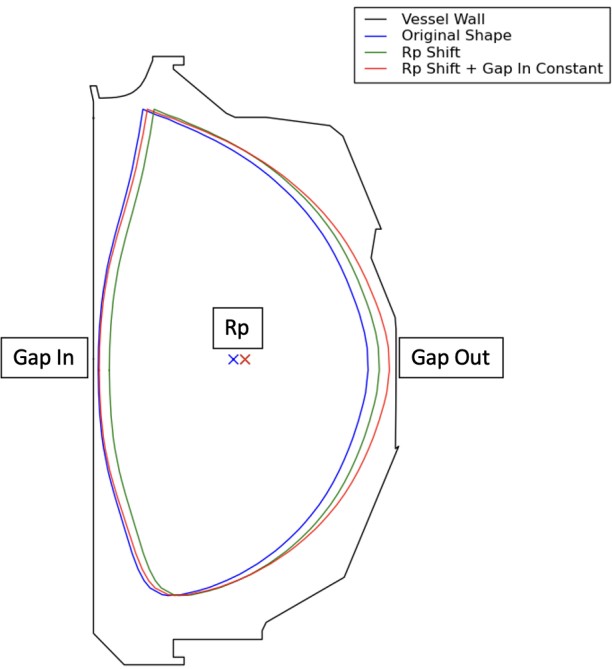

Figure 1: Cartoon diagram showing how the outer wall gap can be adjusted in the Plasma Control System (PCS). Starting with the original shape (in blue), if Rp were increased by rigidly shifting the plasma the outer wall gap would proportionally increase (green). However, the PCS also has other settings like the inner wall gap, so that when exclusively Rp is increased the plasma is also made wider to accommodate maintaining the inner wall gap (blue). In practice, then, increasing Rp by 1 unit while holding all other settings fixed actually decreases the outer wall gap by approximately 2 units rather than exactly 1.

adjust the outer wall gap. What's more, this case was for the DIII-D tokamak where (due to the data sharing policy) open source models are employed: using state-of-the-art models like GPT-4 would likely resolve the minor issues with the generator.

## 4.3 Generic Questions about Tokamak Operations

Unlike previous reactor and experiment-specific use cases, a third possible use related to general tokamak knowledge was explored. To evaluate whether the addition of shot logs through RAG improves GPT-4's responses on general operation of a tokamak, a survey was conducted at a seminar meeting at the MIT Plasma Science and Fusion Center (PSFC) where fusion researchers (ranging from first year graduate students to senior scientists) were asked to choose between responses generated by GPT-4 with and without RAG, anonymized as options A and B. To avoid de-blinding the test with specific shot numbers, we added the following clause to the system prompt "Use both your prior knowledge and information gained from the notes, but please don't directly reference them." The full system prompt can be found in Appendix B.1.

|  | RAG | No RAG |
|---|---|---|
| **Question 1** | 3 | 7 |
| **Question 2** | 3 | 6 |
| **Question 3** | 4 | 5 |

Table 1: Number of fusion researchers who chose answers generated by GPT-4 with and without RAG. The specific criteria was to identify the "most helpful" response.

While the sample size is not large enough to draw rigorous statistical conclusions, the results show that the available results do not reject the null hypothesis that RAG does not improve GPT-4 at answering

Table 2: Question 1 from PSFC Testing: "There is a lot of sputtering off the ICRF antenna; what to do about it?"

| GPT-4 + RAG | GPT-4 |
| --- | --- |
| To mitigate the issue of sputtering from the ICRF antenna, there are several possible approaches that can be applied:

1. Reducing Antenna Power: High power in the ICRF antenna can lead to increased sputtering. Lowering the output power of the antenna, if the operation allows for it, could reduce this effect.

2. Adjusting the Timing: Playing around with the timing when the antenna kicks in may lead to different plasma conditions, potentially reducing the sputtering effect.

3. Changing the Antenna Position: Adjusting the position of the ICRF antenna could have an impact on its operation and lessen the amount of sputtering.

4. Modulating the Frequency: Changing the operating frequency or applying frequency sweep of the ICRF may affect the plasma-antenna interactions and can lead to reduced sputtering.

5. Improving Impedance Matching: An effective impedance match allows maximum power transfer from the source (ICRF system) to the load (plasma). If the matching is not optimal, it can lead to unwanted effects like sputtering. Therefore, working on the impedance match can be a solution. | Sputtering of the Ion Cyclotron Range Frequency (ICRF) antenna in a tokamak is typically an issue related to high power and prolonged plasma operations, which cause the antenna to effectively erode. To mitigate this, there are several possible actions you could take:

1. Operate at lower power: By reducing the power level of the ICRF antenna, you can minimize the heating effects that cause sputtering. However, this might not be ideal as it would also mean less heating of the plasma.

2. Improve the design of the antenna: Sputtering can be reduced by improving the design of the antenna. This includes focusing on the material selection for the antenna, its geometric shape, as well as the surface treatment of the antenna, which can cause it to sputter less.

3. Improve impurity control: Impurities are a primary cause of sputtering. Thus, maintaining effective impurity control within the plasma chamber can make a significant difference in reducing the sputtering issue.

4. Modify the plasma edge conditions: The edge plasma conditions influence the interactions between the plasma and the ICRF antenna. Therefore, manipulating these conditions might be useful to alleviate sputtering.

5. Use protective coatings and limiters: Another approach is to coat the ICRF antenna with a sputter-resistant material or use limiters to protect the antenna from high energy plasma particles. |

generic questions about tokamak operations. This is of particular interest as it indicates that future foundation models trained on a corpora of public data may be helpful even without additional domain specific data provided. Question 1 and the two responses are shown in Table 2, while the other questions and generated responses are listed in the appendix.

One challenge of performing a blind test to measure the benefits of RAG is that questions where RAG shines (the previous two use cases in subsection 4.1 and subsection 4.2) also unblind the test by referencing specific shot numbers and knowledge not in the public dataset GPT-4 was trained on. Thus, our blind test was restricted to relatively generic questions about tokamak operations.

# 5 Discussion

**Future Work**  While leveraging shot logs is a crucial first step for making a tokamak co-pilot, we would also like to note that there are other data sources that one could incorporate to enhance the co-pilot beyond this initial version. For example, we could also enable the source code in the Plasma Control System (PCS) to be queried. Systems such as GitHub Copilot [Chen et al., 2021] have already shown that pairing LLMs with code bases results in powerful tools. In our setting, access to source code could enable operators to quickly diagnose and debug software problems during experiment sessions should they arise.

We also could use more advanced schemes for combining LLM generation and information retrieval. These include having the LLM generate SQL for queries, a map-reduce style architecture for summarizing larger context, and self-introspection in order to determine whether an answer has been found. When deployed on the Discord, 3 out of 7 questions inquired about quantitative information that could have been answered by a reasonable SQL query.

Perhaps the richest source of information not being considered at the moment is the diagnostic data measured for each shot. This data set includes rich, high-fidelity information about the plasma (e.g. temperature, density, and rotation) Stillerman et al. [1997], and has already been leveraged by prior works to predict the evolution [Char et al., 2023][Seo et al., 2023] and stability [Rea et al., 2019][Fu et al., 2020]At the same time, previous works, such as CLIP Radford et al. [2021], have achieved impressive results by learning a mapping between the latent encodings for different modalities of data. Building on both areas of works, we hope to incorporate diagnostic information into our current system. Not only do we expect this to provide additional information and lead to more intelligent answers, but doing so could potentially lead to valuable new use cases for our system such as automatic shot log generation or predicting the evolution of the plasma from text descriptions for scenario development. Any of these could prove valuable to operators and scientists under the extreme attention demands of a tokamak experimental session.

**Conclusion**  In this work we made the first steps towards creating an AI co-pilot and search system for the DIII-D and Alcator C-Mod tokamaks. While there remain several exciting additions that could be made to our system, we believe that our system is already in a state such that it could meaningfully assist physics operators during experiments. We would like to emphasize that the tools used to build this were off-the-shelf, easy to use, and not specific to the particular tokamaks at hand. As such, we believe similar systems could easily be built for other tokamaks or, more generally, scientific communities with access a data set of past experiments. We hope that this work inspires the creation of other co-pilots to aid in scientific discovery.

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

## A Deployment

A bot was created to directly interface with the chatbot in the official DIII-D operations communication software (Discord). Physics operators put in the following 7 questions that day we presented it to General Atomics:

1. Who should I select as Physics Operator?
2. What percentage of discharges in DIII-D ended in a disruption?
3. Who are the top 10 people involved in experiments on DIII-D since it started running?
4. Give me a list of shots that are Super H-mode or SH plasmas or with SH pedestals
5. What is the average delay time reported per experiment in 2022 and 2023?
6. What are some shots that are H-mode, lower single null and detached?
7. What are some shots that had a torque scan and are lower single null or LSN?

## B Prompts

### B.1 System prompt

You are a helpful plasma physics assistant that will try to help out operators of a tokamak by answering their questions using historical notes of tokamak shots. We will present you with a question and a relevant list of shot notes. You will need to use ideas from the results and explain what information you're using to answer the question. Please first generate text thinking it through before starting your actual answer with "ANSWER:".

### B.2 User prompt

Retrieved shot notes: {results}

Question: {question}

# C MIT PSFC Testing Questions 2 + 3

Table 3: Question 2 from PSFC Testing: "What are some causes of disruptions on Alcator C-Mod that are less common on other machines?"

| GPT-4 + RAG | GPT-4 |
|---|---|
| On Alcator C-Mod, several specific factors can potentially cause disruptions that are less common on other tokamaks. These include: 1. Challenges with the control system, such as dealing with flux loop faults, leading to a lack of position control. 2. Certain tokamak conditions can result in disruptions. For example, difficulties are observed during certain transitions, such as during the startup phase or as it transitions into H-mode. 3. Conditions specific to this machine can lead to disruptions. For example, a high-density scan may induce disruption. Similarly, disruptions can occur if certain features of the tokamak (like the A-coils) are not enabled. 4. Nuclear Fusion processes unique to this machine, like locked modes resulting from density transients, could cause disruptions. 5. Hardware issues, such as problems with certain data recording tools like the Aurora 12, that produce inaccurate readings leading to disruption. 6. The coupling on certain parts of the machine (like the 6Bot) can instigate disruptions. Therefore, it is a combination of specific machine set-up, experimental conditions, operational variables and hardware issues that can lead to disruptions on Alcator C-Mod that are less common on other machines. | Some relatively unique causes of disruptions on the Alcator C-Mod tokamak compared to other machines may include the following: 1. High Magnetic Field: C-Mod operates at high magnetic fields, compared to other machines. Disruptions could be caused by miscalculations or mismanagement of these strong fields. 2. High Pressure: The high-pressure plasmas in Alcator C-Mod result in high power densities, which can also lead to disruptions. 3. Compact size: Alcator C-Mod's relatively compact size may be a contributing factor, as it means the plasma is in close proximity to the machine walls. This could lead to higher disruption probabilities when compared to larger machines. 4. Advanced tokamak regimes: C-Mod often operates in advanced regimes which can involve ELM-suppression and high normalized pressures. While these operational modes are desirable for good confinement and higher performance, they do carry inherent risks of disruption. 5. Impurity handling: C-Mod is unique in its use of a high-Z metal like molybdenum as a first-wall material, which might lead to disruptions because of impurity-related causes. 6. Lower toroidal field ripple: A lower toroidal field ripple can lead to disruptions. Few other machines follow parallel operational strategies, making this a unique condition to the C-Mod tokamak. It should be noted that while these causes are less common on other machines, that doesn't mean disruptions on Alcator C-Mod are frequent; they are still generally a rare occurrence. |

Table 4: Question 3 from PSFC Testing: "We are having trouble inducing I-Mode on Alcator C-Mod. Can you provide suggestions on how to access I-Mode?"

| GPT-4 + RAG | GPT-4 |
|---|---|
| To access I-mode in Alcator C-Mod, you might want to explore the following ways: 1. Power and Heating: I-Mode can be induced by injecting enough power into the plasma. This could typically involve high powered RF Heating with over-threshold powers for L-H transition. Please ensure that the auxiliary heating systems such as ICRF are working optimally. 2. Magnetic Configuration: Careful tuning of the plasma's magnetic configuration and operating parameters can increase the likelihood of accessing I-Mode. Check the divertor configuration and edge magnetic shear as these can affect mode access. 3. Plasma Density: Modulating the plasma density to maintain it in an optimum range can facilitate the I-Mode transition. Experiment with different density levels and observe their impact. 4. Pulse Length: In some conditions, increasing the length of the RF pulse can also aid in achieving I-Mode. Please note that even though the above factors can contribute to accessing I-mode, achieving a stable I-mode requires careful balancing of these factors, and might also be dependent on other specifics of your device and operation conditions. | Accessing I-Mode or Improved Mode on the Alcator C-Mod, or any tokamak, usually requires adjusting several key parameters. Specifically, you should consider increasing the power input, while keeping the density down. A relatively higher power-to-density ratio is usually conducive to I-Mode. Upper single-null divertor configurations have shown to be beneficial for reaching I-Mode. Edge localized modes (ELMs) may also play a role in accessing I-mode, as they can help to prevent impurities from accumulating in the plasma. Manipulating magnetic field topology, such as the magnetic shear, might also have an impact. Judicious application of radiofrequency (RF) heating could additionally stimulate transitions to I-Mode. Please remember, these strategies should be deployed subject to the consideration of machine limits to ensure safe operation. As always, monitoring the plasma conditions is paramount for successful operation. |

