# OpenReview forum: "Towards LLMs as Operational Copilots for Fusion Reactors"
_NeurIPS.cc/2023/Workshop/AI4Science — NeurIPS2023-AI4Science Poster_

### Official Review · Reviewer_QXrK · 2023-10-08

**Rating:** 6
**Confidence:** 4

**Review:**

This paper presents a novel approach to assist operators of tokamaks, devices crucial to nuclear fusion research. It introduces a retrieval augmented generation system that leverages text logs generated during experiments.

Strengths:
The problem is very novel.
The paper's focus on practical application, such as assisting with device-specific operations, addresses real-world challenges faced by operators in tokamak experiments.
The authors provide transparency by detailing the datasets and methodology used to create the copilot, enhancing the reproducibility of the research.

Weaknesses
The major concern is that there is no quantitative evaluation of the proposed framework. Only qualitative evaluation and examples, such as prompts and sample generated texts are shown. It would be great if you can add some numerical results to demonstrate the usefulness of your framework and the problem itself.

---

### Official Review · Reviewer_njDF · 2023-10-25
**This research uses LLMs to act as copilot for tokamak experiments action through direct generatin and RAG.**

**Rating:** 4
**Confidence:** 4

**Review:**

The application to tokamak is exciting, but the current work did not provide more valuable contributions like data or models to the community. Besides, the human evaluation is only limited to very limited cases, making the results not convincing enough.

Also, the analysis of results is very limited. I am expecting a more detailed comparison of different methods.

---

### Meta-Review · Area_Chair_bTuF · 2023-10-27

**Recommendation:** Accept (Poster)
**Confidence:** 3

**Metareview:**

The authors consider a new way to obtain helpful text responses from different LLMs by augmenting it with text logs from tokamak operators through retrieval from a vector database, in a way which might be useful in contexts beyond tokamak operation. They provide qualitative results and comparison to a baseline without RAG, and do not describe any quantitative results or metrics which might allow more careful evaluation or comparison, but the paper will be a source of useful discussions.